# SpectroBank: A filter-bank convolutional layer for CNN-based audio applications

## Abstract

We propose and investigate the design of a convolutional layer where kernels are parameterized functions. This layer aims at being the input layer of convolutional neural networks for audio applications. The kernels are defined as functions having a band-pass filter shape, with a limited number of trainable parameters. Building on the literature on this topic, we confirm that networks having such an input layer can achieve state-of-the-art accuracy on several audio classification tasks. We propose new filter functions and explore the effect of different parameters on the network accuracy and learning ability. This approach reduces the number of weights to be trained and enables larger kernel sizes, an advantage for audio applications. Furthermore, the learned filters bring additional interpretability and a better understanding of the data properties exploited by the network.

## 1 Introduction

In audio signal processing, time-frequency representations such as spectrograms are central tools. They have an intuitive interpretation and reveal insightful information to the human expert. It is not a surprise that many deep learning approaches to audio signals use such representations as well (Choi et al., 2017; Purwins et al., 2019). It is also convenient as most of the deep network architectures have been developed for image processing and require 2D arrays of values as inputs. The network learns to detect time-frequency patterns, similarly to what is done on images. Depending on the task, it may then output a classification of a sound (Piczak, 2015; Salamon & Bello, 2017), a denoised signal (Lu et al., 2013) or separated sources (Chandna et al., 2017).

However, natural images and spectrograms do not possess the same properties and turning an audio file into an image has some limitations. Among them, spectrogram representations can be defined in many different ways, with different time window shapes and sizes or different frequency spacing. Choi et al. (2017) and Purwins et al. (2019) give a review of the different time-frequency representations used in deep learning. In addition, patterns in the time-frequency plane are different from those that can be found in images: the former are usually less complex, with smoother edges and limited textures. Furthermore, the axes are not equivalent in the spectrogram as frequency is different from time. For example a frequency-shifted pattern may result in a different sound classification (Lee & Nam, 2017), while a temporal shift does not. Moreover, the spectrogram is the magnitude of the short-time Fourier transform and the information contained in the phase is not taken into account. Lastly, computing a spectrogram, and possibly inverting it for synthesis, adds a computational burden which can be important for large audio datasets.

To overcome these limitations, an alternative direction has been chosen consisting of taking an end-to-end approach where the raw audio file is the input of the network. The recent success of Wavenet (Oord et al., 2016; Paine et al., 2016; Oord et al., 2017) demonstrates the efficiency of this approach for audio synthesis. Raw audio input is also beneficial for speech separation tasks. Tasnet (Luo & Mesgarani, 2018) as well as Wave-U-Net (Stoller et al., 2018) show better performances for speech separation and faster processing compared to spectrogram-masking approaches.

In end-to-end approaches, one-dimensional convolutions are applied to raw audio signals. However, kernel size needs to be much larger than the one used for image applications. Indeed, at a sampling rate of 44kHz, 44 samples represent 1 ms of audio signal. To capture audio patterns that have duration of 10, 100 ms or more, in particular low frequency patterns, either large kernels are needed or deeper convolutional architectures (to allow for combinations of kernels at many different positions in time).

Both solutions lead to a large increase in the number of parameters to be learned and hence require more training time and more data. The "atrous" convolution have been introduced in Wavenet in order to increase the time length of the kernel without increasing the number of weights to learn. Finding alternative ways for unlocking the time-length limit is an important challenge for raw audio processing in deep learning.

**Motivation.** In the case of end-to-end learning, several studies investigate convolution kernels learned from the raw audio signal (Dieleman & Schrauwen, 2014; Tüske et al., 2014; Golik et al., 2015; Sainath et al., 2015). They all show that the input kernel's focus in frequency is similar to the one of the Mel or auditory scale. The kernel shapes in the spectral domain are similar to band-pass filters, with more narrow-band kernels localized on the low frequency spectrum than in the high frequency. This behavior does not depend on the network architecture nor on the application such as speech recognition (Hoshen et al., 2015; Zeghidour et al., 2018) or audio tagging (Dieleman & Schrauwen, 2014). All of these results suggest that the logarithmic spacing of frequencies and bandwidth properties first established in the psycho-acoustics studies with the Mel/Bark scales are somehow universal in audio analysis tasks. These works point out the tendency of the input convolution kernels to adopt band-pass filter shapes. Hence, we hypothesize that designing kernels with a band-pass property results in an inductive bias that helps the network converge more rapidly and possibly reduces overfitting. Our first motivation is to confirm this hypothesis.

The studies cited above remain experimental without, yet, precise spectral and temporal properties of the kernels. In addition, most of them initialize the kernels as band-pass filters with a Mel scale frequency spacing. So the influence of the kernel initialization remains unclear. Our second motivation is to investigate more precisely these filters' properties.

Adopting a hybrid approach, halfway between the raw audio and the spectrogram, we propose to learn particular filters' shapes having a limited number of parameters that fully define them. These filters are the kernels of the first convolutional input layer of the network. This set of kernels may be seen as a filter bank. Consequently, the new input layer acts on the raw audio and outputs a learned time-frequency representation, adapted to the task. The functions we propose are modulated Gaussian windows, Gammatone and Gammachirp functions. The goal of these filters is two-fold. Firstly, it reduces the number of parameters to learn. It makes the size of the kernel independent of the number of weights to learn and enables the usage of large temporal inputs. Secondly, this layer of parameterized functions helps understanding the filtering process done within the first layer of deep networks. This opens the way to a better interpretation of the neural networks and beyond, of the intricate relationship and the shape of audio patterns in the time-frequency space.

**Our contribution** [1]. We propose new parameterized functions and compare them to recent works on the same topics that use learnable Wavelets or sinc functions (see *related work* section below). We confirm that this approach improves the accuracy on several audio classification tasks. Moreover, combining the Spectrobank layer with simpler networks can lead to a higher accuracy. We explore the influence of different parameters on the learning, such as the numbers of kernels and their length. Our classification experiments show that the number of filters required to obtain the best results remains small, around 20-30. We also demonstrate that the performances of different functions proposed in audio signal processing (modulated Gaussian, Gammatone and Gammachirp functions) give close results and are better than Wavelets at classifying sounds. Last but not least, a relationship between the central frequency of the filter and its temporal width emerges with the learning. We provide evidences that the network converges to an auditory frequency spacing, close to the ERB (Equivalent Rectangular Bandwidth) and Bark scales found in psycho-acoustic studies.

**Related work.** The kernel shapes proposed in the present work are based on specific signal processing functions. They are used for performing short-time Fourier transforms or more generally for designing filterbanks. Modulated (truncated) Gaussian are emblematic examples. Gammatones and Gammachirp functions are used in cochlear models (Saremi et al., 2016). They provide interesting results when combined with deep learning models for speech enhancement (Baby & Verhulst, 2018).

Learning parametric filters is halfway between 1) learning a standard convolutional layer, where all the weights of the kernels are learnable and unconstrained and 2) having a layer of kernels being fixed functions, where only the combination of these predefined functions may be learnt. The first approach is the most versatile but is more prone to overfitting. The second approach used for example in the

---

[1]The SpectroBank code will be made publicly available after the review process.

Scattering transform (Bruna & Mallat, 2013; Andén & Mallat, 2014; Cotter & Kingsbury, 2019), or in Jacobsen et al. (2016) benefits from an inductive bias through the chosen kernel functions but is less flexible. The concept of learning filters aims at making an ideal compromise between flexibility and inductive bias. It has been first introduced in three recent works by Seki et al. (2017), Ravanelli & Bengio (2018) and Khan & Yener (2018). The first one introduces Gaussian filters in the input layer. Parameters are the amplitude, the Gaussian width and the modulation frequency. An increase of the classification accuracy is reported with the learned parameters. However, the filter learning is seen as a fine-tuning of the network after the first training pass with fixed Gaussian parameters. In the present work, the filter layer is fully integrated in the learning process, the parameters are learned from the beginning. In Ravanelli & Bengio (2018), the authors introduce a layer, called *SincNet*, made of sine modulated functions that approximate band-pass rectangular windows in the frequency domain. The learned parameters are the minimal and maximal cut-off frequencies of each band-pass filter. One of the main results is given by the cumulative frequency response of the SincNet filters. The network tends to focus more on particular regions of the frequency space, where formants are localized. This is interesting as it shows how the parameterized filters enable a precise interpretation of the learning and underline particular spectral properties of the data. The present work goes further in this direction. Eventually, Khan & Yener (2018) introduce Wavelet filter banks learned for speech recognition. Each kernel is a Wavelet defined by a single parameter, its scale. It shows evidences both of the efficiency of this approach and of the possibility to interpret the shape of the learned kernels. We compare the efficiency of the Wavelet filters with several other modulated windows and show that the former under-performs on audio signals. More recently and in line with our approach, Loweimi et al. (2019) present complementary results, on a different dataset, with a focus on the sinc-square function, learning either the frequency or the bandwidth of the filters.

## 2  LEARNABLE FILTER BANKS (SPECTROBANK)

We design a new convolutional neural network layer, called SpectroBank. In this layer the kernels are functions defined by a few parameters that are learned. We call these functions *filters*, making a parallel with filters in signal processing. Indeed, these functions have the property of being band-pass filters and are well known in audio signal processing. One of the trainable parameters of each filter is the central frequency of the band-pass filter. The second parameter is the bandwidth of the filter (or a quantity closely related to it). Hence this set of filters forms a filter bank where the frequency and bandwidth of the filters may be adapted to the data and to the learning task. Note that the learned filterbank may not cover the entire spectrum but should focus on important spectral regions that are the most discriminative for classification.

The input of the SpectroBank layer is a 1D audio signal and the output is a 2D representation. The output representation axes are time and filter number. Since each filter is associated to a particular frequency band, this 2D representation can be seen as a time-frequency one (or time-scale in the case of Wavelets). Initializing the filters by increasing frequencies (or scales), we can influence the frequency ordering to follow the filter number.

In all the definitions, $N$ denotes the filter length and $n$ is the variable (sample number). The time in second can be expressed using the sampling frequency $f_s$ with $t = n/f_s$ and the frequency in Hertz with $f \times f_s$, where $f \in [0, 0.5]$ is the normalized frequency in the formulas.

**Mexican hat Wavelet**. In order to compare to the state-of-the-art, we use the Mexican hat Wavelet introduced in the paper by Khan & Yener (2018):

$$w(n) = \frac{2}{\pi^{1/4}\sqrt{3s}} \left( \frac{n^2}{s^2} - 1 \right) e^{-\frac{n^2}{s^2}}, \tag{1}$$

with $n \in [-N/2, (N-1)/2]$ and $s > 0$ being the scale parameter.

**Gaussian filter**. Here, $n \in [-N/2, (N-1)/2]$. The Gaussian filter $g$ is defined as follows:

$$g(n) = \sqrt{\frac{2}{\sqrt{\pi}\sigma}} e^{-\frac{n^2}{2\sigma^2}} \left( \cos(2\pi f n) + i \sin(2\pi f n) \right). \tag{2}$$

The parameter $\sigma > 0$ is the variance of the Gaussian (temporal window width) and $f$ is the oscillating frequency. It is a complex-valued function that we split into its real and imaginary parts. For each $f$ and $\sigma$ two kernels are created, one with the cosine modulation and one with the sine one.

**Gammatone filter**. The Gammatone filter (Darling, 1991; Patterson et al., 1992; Hohmann, 2002) is another example of kernel. It is defined on the interval $n \in [0, N-1]$ as :

$$h(n) = A(\gamma, b)n^{\gamma-1}e^{-2\pi bn}\left(\cos(2\pi fn) + i\sin(2\pi fn)\right), \tag{3}$$

where $A$ is the normalization, $A(\gamma, b) = \sqrt{2(4\pi b)^{(2\gamma+1)}/\Gamma(2\gamma+1)}$. The parameter $\gamma$ is the order of the Gammatone. It can be learned or fixed to 2 or 4. These two orders are the best suited ones for modeling the human hearing related filter bank (Patterson et al., 1987). In the experiments, we will fix $\gamma = 2$ or $\gamma = 4$. The other learnable parameters are $b$, related to the width of the function, and $f$ the frequency. The symbol $\Gamma$ denotes the Gamma function. The bandwidth $B$ of $h$ depends linearly on $b$ and is given by the following formula (Darling, 1991):

$$B(\gamma, b) = 2(2^{1/\gamma} - 1)^{1/2}b. \tag{4}$$

**Gammachirp filter**. This function is similar to the Gammatone family ones but possesses an oscillating frequency that may evolve with time. The Gammachirp function (Irino & Patterson, 1997) is defined on the interval $n \in [0, N-1]$ as follows:

$$k(n) = A(\gamma, b)n^{\gamma-1}e^{-2\pi bn}\left[\cos(2\pi fn + c\ln(n+\epsilon)) + i\sin(2\pi fn + c\ln(n+\epsilon))\right], \tag{5}$$

where $A$ is defined above. In the present work, $\gamma$ is fixed to $\gamma = 4$. This filter possesses 3 parameters, $b$ related to the width of the window, $f$ to the frequency and $c$ to the chirp value. To avoid the logarithmic singularity at the origin, we add a small positive value $\epsilon = 10^{-4}$ to the expression.

*Remark 1*: All the functions are defined and normalized in the continuous domain. In our application, the filters are discretized and truncated in order to be implemented in the convolution layer. Since they all vanish away from zero, it remains a good approximation, provided that the function's width does not exceed the fixed filter length $N$.

*Remark 2*: The modulated window functions are defined with a cosine (real part) and a sine (imaginary part) term, relating them to the Fourier transform, the spectral domain and the standard definition of filters in signal processing. For the sake of simplicity, in our experiments, we have chosen to use only the cosine term. The absence of the sine term did not affect the accuracy of our classification results. The network is able to adapt and detect discriminative patterns with a shifted cosine modulation.

*Remark 3*: It is important to distinguish the filter length $N$ from the filter temporal width $\sigma$ or $b$ (or $s$ for the scale). The filter length is fixed, can not be learned and is the size of the vector on which the filter is defined. The temporal width is learned and specifies the spread of the function over the vector of size $N$. Therefore, the filter temporal width is always smaller than the filter's length.

## 3 EXPERIMENTS AND RESULTS

We apply SpectroBank to several classification tasks described in the following sections. We want to assess it on standard tasks found in the literature presented in the introduction. We have chosen 2 freely available speech datasets: *AudioMNIST* (Becker et al., 2018) and *Google Speech Commands v2* (Warden, 2018). Both datasets contain words pronounced by different speakers. These datasets are dedicated to limited-vocabulary speech recognition tasks and the goal is to train the network to correctly recognize the word present in each audio sequence. We also investigate the performances of SpectroBank on an environmental sound dataset in order to cover more diverse audio patterns. We have chosen the UrbanSound8K dataset (Salamon et al., 2014). This dataset have been used recently for end-to-end learning (Dai et al., 2017; Abdoli et al., 2019). Statistics and spectral energy distribution per class of these datasets are given in the Appendix C.

In order to compare the impact of the SpectroBank layer on the learning and classification results, we use existing network architectures and modify the first layer. For networks with raw audio input, the first convolutional layer (performing a standard 1D convolution) is replaced by our proposed parameterized convolution layer. Our layer is then followed by a non-linear ReLU activation function. A stride parameter is available allowing to define the overlap in time of consecutive convolutions. Our focus being to learn from audio, we decided to compare our approach only to similar techniques, despite the fact that image-based network achieve sometimes higher accuracy than the purely audio-based ones. All the models used for the experiments were implemented using the Keras framework

and will be made publicly available along with the final version of the paper. Detailed architectures of all networks can be found in appendix B. Training was performed using a NVidia GTX1080Ti having 11 GB of RAM.

**Input layer initialization**. When initializing a filter bank for learning, most of the available solutions start from a filter bank with a Mel-scale (or log-scale) frequency spacing (Sainath et al., 2013; Zeghidour et al., 2018; Ravanelli & Bengio, 2018; Khan & Yener, 2018). This frequency distribution is supposed to be optimal for audio processing and learning. However, in the present work, we want to check this assumption. Hence in all the experiments (except when comparing with SincNet where we retain the mel-based initialization from Ravanelli & Bengio (2018)), we use linearly spaced frequencies (or scales), distributed over the entire spectrum and a constant initialization value for the bandwidth ($\sigma$ or $b$).

### 3.1 THE IMPACT OF SPECTROBANK PARAMETERS

The learnable parameters of the SpectroBank filters are not the only values that may influence the network accuracy. The choice of the filter type is important as well as the filter length and the layer stride (filter overlap). We have tested different configurations and the results are shown on Fig. 1. On the left, the accuracy increases with the number of filters up to around 30. Beyond this, no improvement is reported. This number is hence a good compromise between accuracy and network complexity. Similar trend holds for all filter types. These results highlight the better performance of the modulated windows compared to the Wavelets. On Fig. 1b, the impact of the filter overlap

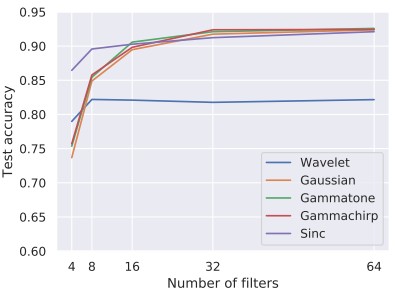

(a) Accuracy for different filter types and numbers of filters. The modulated windows filters achieve similar performances and reach a better accuracy than the Wavelets.

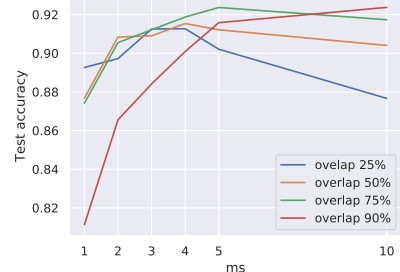

(b) Accuracy of Gammatone filters for different lengths and overlap ratios.

Figure 1: Influence of several SpectroBank layer properties on the network accuracy. (Dataset: Google Speech Commands)

(or kernel stride) is shown, exhibiting two different behaviors. First, for large windows (beyond 5ms), a large stride lead to a drop in accuracy. Indeed, the filter width (spread of the modulated window or Wavelet) may be much smaller than the filter total length $N$. Nevertheless, the overlap is measured on the total length. Narrow windows may not overlap at all and information is lost during the convolution process. Secondly, short kernels (less than 4ms) with large overlap (or small stride), can render the network short-sighted in time. In that case, long temporal patterns require the combination of a large amount of successive output values. The convolutional layers following the SpectroBank layer, deeper inside the network, may not be able capture these long patterns. This results as well in a drop of the accuracy observed on Fig. 1b.

### 3.2 AUDIOMNIST RESULTS

The original AudioMNIST paper (Becker et al., 2018) performs digit classification using raw audio as input to a network called AudioNet. The code[2] supplied with the paper has been re-used to perform 5-fold validation on the data. AudioNet is made of six convolutional layers, each convolution

---

[2]https://github.com/soerenab/AudioMNIST

being followed by a max-pooling layer, and two dense layers, connected to an output layer. In all tests performed using this dataset, the models were trained using the Adam optimizer with default parameters during 50 epochs. Batch size used was set to 256 and loss function used was the categorical cross-entropy. Test accuracy was then computed after this training phase and the same process was repeated for each fold.

On the AudioMNIST dataset sampled at 8 kHz, AudioNet has ca. 17 million trainable parameters. The original paper from (Becker et al., 2018) claims an accuracy of $92.53\% \pm 2.04\%$, whereas our implementation of AudioNet using Keras and Adam optimizer (instead of SGD in the original paper) yields an average accuracy of $94.9\% \pm 1.54\%$, which is already a significant improvement. We performed the same 5-fold validation using a modified version of AudioNet where the first convolutional layer is replaced by a *SpectroBank* layer. This layer consists in 32 4th-order Gammatone filters of length 80 (corresponding to 10 ms at 8 kHz). The stride has been set such that the overlap between two consecutive convolution steps is equal to 75%. In this modified network, the number of trainable parameters drops to ca. 3.5 million trainable parameters, i.e. a reduction in size by a factor 5. Using the SpectroBank-enabled AudioNet the average accuracy increases to $96.8\% \pm 1.22\%$.

Another SpectroBank-enabled network was used to perform the classification task on AudioMNIST. The architecture, loosely adapted from the one used in the paper by Abdoli et al. (2019), is described in appendix (Table 5). Despite its much smaller number of trainable parameters (ca. 300'000), its average accuracy improves to $98.0\% \pm 0.41\%$. For the sake of completeness, we also trained this network, replacing the Gammatone filters by the learned wavelets as in (Khan & Yener, 2018), and the learned SincNet filters from (Ravanelli & Bengio, 2018). A summary of all results achieved using AudioMNIST can be found in Table 1.

Table 1: AudioMNIST mean test accuracy

| Network | # Trainable parameters | Avg. accuracy |
|---|---|---|
| AudioNet | 17 M | $94.9\% \pm 1.54\%$ |
| SpectroBank-AudioNet | 3.5 M | $96.8\% \pm 1.22\%$ |
| **SpectroBank-custom (Gammatone)** | **300 k** | **$98.0\% \pm 0.41\%$** |
| SpectroBank-custom (SincNet) | 300 k | $97.2\% \pm 1.0\%$ |
| SpectroBank-custom (Wavelet) | 300 k | $89.9\% \pm 1.18\%$ |

## 3.3 GOOGLE SPEECH COMMAND RESULTS

The Google Speech Command dataset provides similar data to the AudioMNIST one, with a larger number of classes (35) to distinguish. In the original setting, the goal was to classify 15 unwanted words together as *unknown*. However, in the experiments we performed, we classify each word independently. This dataset does not have pre-defined folds, but train, test and validation data are specified explicitly. We focus on the "*Basic*" network of the *SampleCNN* group described in Kim et al. (2019). Using an input signal resampled to 22.05 kHz, the Basic network has 8 identical blocks, each block being made of a 1D convolution (size 3), followed by a batch normalization, ReLU activation and max pooling. In our experiments, we adapted the proposed setting in order to keep the original 16 kHz sampling of the dataset and ended up with a 7 blocks (vs. 8) network in order to avoid empty dimensions. The code[3] provided by Kim et al. (2019) was used as basis for our experiments. Reducing the number of blocks to 7 and keeping the original 16 kHz sampling rate yields networks having similar number of trainable parameters (ca. 2.3 million vs ca. 2.5 million respectively for 7 blocks/16 kHz and 8 blocks/22.05 kHz).

Given that Google Speech Commands does not possess pre-defined folds for $n$-fold validation, the experiments were repeated 5 times in order to compute the mean accuracy. The original results from Kim et al. (2019) give an average accuracy of $92.5\% \pm 0.7\%$ (averaged over 3 training runs). When reproducing a similar experiment (training performed with SGD optimizer, with early stopping), with the simplified SampleCNN using 16 kHz data, we found the average accuracy to be $93.34\% \pm 1.26\%$.

---

[3]https://github.com/tae-jun/sampleaudio

We created a SpectroBank-enabled version of SampleCNN, replacing the first block by a spectrobank layer and modifying the other basic blocks introduced by Kim et al. (2019), as described in appendix B, table 8. The SpectroBank layer is made of 80 order-4 Gammatone filters, overlapping by 80% and having a length representing 10 ms. As our initial layer contains less filters than the initial implementation (80 vs. 128), the basic block modifications allow to keep non-empty sizes when the number of basic blocks increases. The number of basic blocks is identical (7), reducing the number of trainable parameters to 1 million. Unlike the original paper, this network was trained using the Adam optimizer, while keeping the same learning rate reduction strategy. The early stopping is usually activated after less than 20 epochs. The mean accuracy achieved using this network improves slightly to $93.45\% \pm 1.35\%$.

## 3.4 URBANSOUND8K RESULTS

One of the main interests of this dataset resides in the fact that the environmental sounds exhibit spectral characteristics that are quite different from speech datasets studied in the previous sections. It is however a more challenging dataset, firstly because its size is almost an order of magnitude smaller, and secondly because of the longer input data (each sample being 4 seconds long).

We base our experiments on the works from Dai et al. (2017) and more recently Abdoli et al. (2019), that also perform classification task using convolutional networks on raw audio input. Dai et al. (2017) define several network architectures, with numbers of trainable parameters ranging from 200'000 to 4 millions. We will focus on the two smallest networks, referred to as M3 and M5 in the original paper. Despite dataset being split into 10 folds for training and validation, only one test (using the 10th fold for validation) has been done in Dai et al. (2017). We tested those networks using an existing Keras implementation[4] and performed 10-fold validation to get the mean accuracy over all folds, using data resampled to 8 kHz. The average accuracy for M3 was found to be $58.94\% \pm 3.83\%$ (vs. 56.12% in the original paper) and the one for M5 $66.98\% \pm 6.37\%$ (vs. 63.4% initially).

SpectroBank-enabled versions of M3 and M5 have been created for comparison. The first layer consists in 24 4th-order Gammatone filters, overlapping by 75% and having a length representing 10 ms. All networks were trained for 100 epochs using the Adam optimizer, reducing the learning rate by a factor 2 after 10 epochs without improvement of the validation loss. SpectroBank-enabled M3 accuracy is very close to the one achieved with initial M3, namely $59.17\% \pm 5.33\%$. However, the SpectroBank-M3 has ca. 22'000 parameters, i.e. close to ten times less than initial M3. In the case of SpectroBank-M5, mean accuracy is improved to $67.45\% \pm 5.48\%$ (with a number of trainable parameters very close to initial M5, i.e. slightly more than 500'000). We also tested the SpectroBank-SampleCNN architecture described in section 3.3, and achieved a mean accuracy of $69.16\% \pm 5.95\%$. When comparing more specifically the 10-th fold best accuracy achieved by Dai et al. (2017) is 71% using M18 model (3.7 million parameters), while our approach reaches an accuracy of 75.8%. Higher accuracies have been achieved on this dataset using raw audio (Li et al., 2018), they however resort to data augmentation, which was not used in our experiments. All results are summarized in table 2.

Table 2: UrbanSound8K mean test accuracy

| Network | # Trainable parameters | Avg. accuracy |
|---|---|---|
| M3 | 222 k | $58.94\% \pm 3.83\%$ |
| **SpectroBank-M3** | **22.5 k** | **$59.17\% \pm 5.33\%$** |
| M5 | 561 k | $66.98\% \pm 6.37\%$ |
| **SpectroBank-M5** | **513 k** | **$67.45\% \pm 5.48\%$** |
| **SpectroBank-SampleCNN** | **1 M** | **$69.16\% \pm 5.95\%$** |

The approach taken by Abdoli et al. (2019) is to perform classification on overlapping splits of initial audio data (usually having a length of 1 second). They however also compare to a network taking a single block of data (having a length of ca. 3 seconds). While the code was supposed to be made

---

[4]https://github.com/philipperemy/very-deep-convnets-raw-waveforms

available after final publication, the repository[5] was still empty. The model was then reimplemented and trained according to the description found in the paper, using all 4 seconds of input data instead of trimming it to 3 seconds. Instead of the mean accuracy claimed ($83\% \pm 1.3\%$, from Table 2 in original paper), our tests only achieved $63.8\% \pm 5.68\%$, which is a significant difference. We have been unable so far to explain this discrepancy.

### 3.5 PERFORMANCE CONSIDERATIONS

While having a smaller number of trainable parameters compared to a classical convolutional layer, the SpectroBank layer is slightly more complex than a simple convolution, as it requires to compute the filter coefficients from the parameters, then perform the convolution. Our implementation showed SpectroBank was roughly $50\%$ slower compared to a convolution layer having the same number of filters. This performance drop is however mitigated by the fact that the bigger stride (usually an order of magnitude bigger) used in the experiments decreases the overall number of convolution operations required, resulting in faster training times, even when using a non-logarithmic or random frequency initialization scheme.

### 3.6 PROPERTIES OF LEARNED FILTERS

The learned parameters of the SpectroBank filters can reveal insights about the data and the learning process. As stated in the introduction, several studies have shown a tendency governing the spacing in frequency of their learned kernels, approximations of band-pass filters. The spacing becomes exponentially large as the frequency increases, following what is called a Mel scale. This is in agreement with psycho-acoustics tests on the human cochlear system. In order to go further in this direction, we investigate 1) the frequency spacing and 2) we test the relationship between the temporal width of the filters and their central frequency. Indeed, psycho-acoustic models (the equivalent rectangular bandwidth (ERB) model (Glasberg & Moore, 1990) and the Bark model (Zwicker & Terhardt, 1980)) provide such a relationship. This is made possible by our approach where the temporal width as well as the filter central frequency are well defined for each filter.

**Frequency spacing**. The SpectroBank layer is initialized with a linear frequency spacing from 0 to the Nyquist frequency. After the learning phase, the filter frequencies have evolved and moved away from their initial value as can be seen on Fig. 2a. The frequency distribution is not exponential but we can point out several interesting facts. Firstly, the final curve is flatter than the initialization in the range 0-2kHz (more filters in this range). It shows that the network tends to favour filters with a band-pass in this range for its discriminative process. Secondly, beyond 4kHz, the filters stay close to their original value. This suggests that there is not enough meaningful information in this frequency range for a correct learning. This is indeed the case for speech where the main information resides below 4kHz (see Appendix C).

**Bandwidth and frequency**. The learned filter banks can be compared to filter banks modeling the human auditory system. Two main models can be found in the literature, the Equivalent Rectangular Bandwidth (ERB) model (Glasberg & Moore, 1990), and the Bark model (Zwicker & Terhardt, 1980). In these models the bandwidth $B$ of the filter is related to its central frequency $f$ by explicit formulas given in Appendix D. The ERB and Bark curves are plotted on Fig. 2b, together with the learned parameters of the Gammatone filters (black dots). We observe a very good agreement between the ERB curve and the learned filters for frequencies below 2kHz. Ravanelli & Bengio (2018) show that for a neural network applied to a speech dataset, the focus of the learning is situated around the pitch frequency located at 130Hz (male) and 230Hz(female), and the first and second formants, which are around 500Hz and 1kHz respectively. This is exactly the frequency region where our learned filters match the ERB scale.

**Cumulative distribution**. In Figure 3 we can see the cumulative energy distribution, in the frequency domain, of the learned filters for Gammatone and Sinc filters . We have used 32 filters during the training. From the Gammatone distribution, we can observe that filters focus on at least some of the frequencies relevant for speech, as discussed earlier, in the range 100Hz - 1kHz. The sinc distribution has the same global shape as in Ravanelli & Bengio (2018) but is less conclusive about the formants.

---

[5]`https://github.com/sajabdoli/Environmental_sound_classification`, last accessed Nov 15th 2019

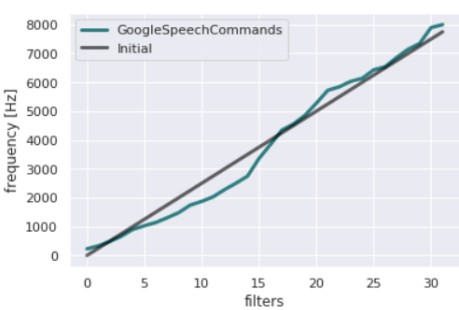

(a) Frequency distribution of the filters before (straight line) and after training (green curve)

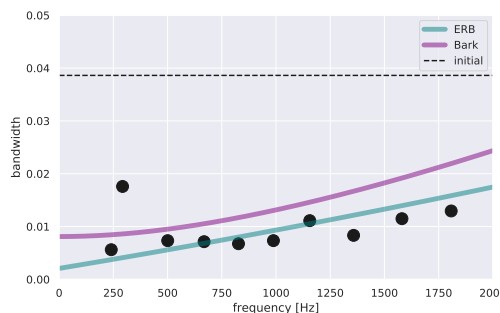

(b) Bandwidth and frequency of the learned filters (black dots) over the range 0-2kHz. The curves are the psycho-acoustical relationships given by the ERB and Bark scales. Black dashed line: initial bandwidth value for all filters.

Figure 2: Bandwidth and frequency of the learned Gammatone filters ($B$ of Eq. (4) and $f$ parameters) using the Google Speech Commands dataset

We also note a difference in the low-frequency region below 100Hz, where the distribution drops in our case. We point out that both our dataset and classification task are different, which could explain the discrepancies. It still shows the high focus of the filters on the range 100Hz - 1kHz, where the distribution curve is the highest.

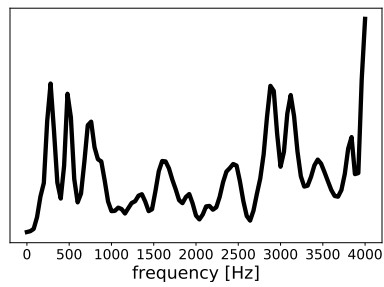
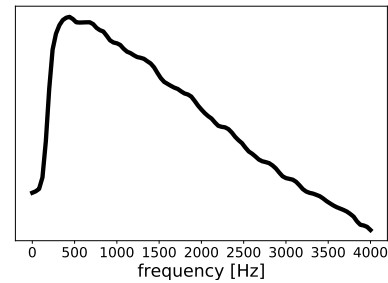

Figure 3: Cumulative frequency energy distribution for learned filters on AudioMNIST dataset, SpectroBank-XS network trained with Gammatone (order 4) and SincNet first layer, with 32 filters.

## 4    CONCLUSION

Decades of research in audio signal processing have brought us an important knowledge about sounds, speech and audio information. This knowledge may be inserted within neural networks as a priori information and turned into efficient inductive biases. This is what we show with the example of the SpectroBank layer, a layer of parameterized filters adapted to the extraction of audio information. Moreover, the trained network possesses properties than can, in turn, bring new insights about audio data back to the audio signal processing community.

Future work in this direction and further developments of convolutions with parameterized functions may lead to important progress both in deep learning and audio signal processing. The reduction of the number of trainable parameters decreases the network complexity, along with the training time. It also enables a better interpretation of the network adaptation to the data.

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

## A    FILTER BANK PARAMETERS

In this section we provide an example of visualization of the filter functions introduced for SpectroBank. These functions and their parameters are recalled on Table 3. Their shape in time is illustrated on Fig. 4, with increasing oscillating frequency (or scale for wavelet) from blue to purple (starting from $f = 0$).

Table 3: Description of the filter bank types and the parameters used during training. In most of our experiments, $\gamma$ is fixed to $4$.

| Filter Type | # of parameters | Parameters |
|---|---|---|
| Wavelet | 1 | $s$ - scaling |
| Gaussian | 2 | $f$ - frequency $\sigma$ - width |
| Gammatone | 3 | $f$ - frequency, $b$ - bandwidth, $\gamma$ - order |
| Gammachirp | 3 | $f$ - frequency, $b$ - bandwidth, $c$ - chirp trend |

## B    NETWORK ARCHITECTURES

Detailed architecture for SpectroBank-enabled networks used in the experiments are given in this section. All convolutional and dense layers use ReLU activation, except for the output layer using softmax.

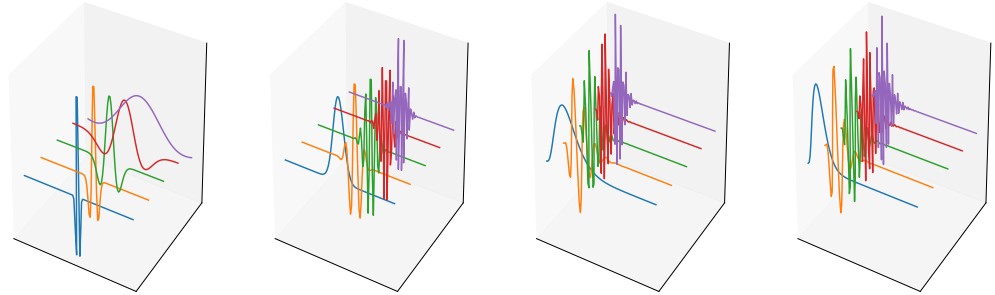

Figure 4: Examples of filter banks in time domain. From left to right: Wavelet filters, Gaussian filters (cosine modulation), Gammatone filters (envelope, cosine and sine modulations) and Gammachirp filters, for fixed bandwidth and different frequencies.

Table 4: SpectroBank custom architecture for AudioMNIST.

| Layer | Output size |
|---|---|
| Input | $8000 \times 1$ |
| SpectroBank (32 filters, size 80, stride 20) | $400 \times 32$ |
| Convolution (32 filters, size 32, stride 2) | $200 \times 32$ |
| MaxPooling (stride 4) | $50 \times 32$ |
| Convolution (64 filters, size 16, stride 2) | $25 \times 64$ |
| Convolution (128 filters, size 8, stride 2) | $13 \times 128$ |
| Convolution (256 filters, size 4, stride 2) | $7 \times 256$ |
| MaxPooling (stride 4) | $1 \times 256$ |
| Dense (128) | 128 |
| Dropout 0.5 | 128 |
| Dense (64) | 64 |
| Dropout 0.5 | 64 |
| Dense 10 | 10 |

Table 5: SpectroBank-XS custom architecture for AudioMNIST.

| Layer | Output size |
|---|---|
| Input | $8000 \times 1$ |
| SpectroBank (32 filters, size 80, stride 20) | $400 \times 32$ |
| MaxPooling (stride 4) | $100 \times 32$ |
| Dense (16) | 16 |
| Dropout 0.5 | 16 |
| Dense 10 | 10 |

Table 6: M3-SpectroBank custom architecture for UrbanSound8K.

| Layer | Output size |
|---|---|
| Input | $32000 \times 1$ |
| SpectroBank (24 filters, size 80, stride 20) | $1600 \times 24$ |
| Batch Normalization | $1600 \times 24$ |
| MaxPooling (stride 4) | $400 \times 24$ |
| Convolution (256 filters, size 3, stride 1) | $400 \times 256$ |
| MaxPooling (stride 4) | $100 \times 256$ |
| Global Average Pooling | $256$ |
| Dense 10 | $10$ |

Table 7: M5-SpectroBank custom architecture for UrbanSound8K.

| Layer | Output size |
|---|---|
| Input | $32000 \times 1$ |
| SpectroBank (24 filters, size 80, stride 20) | $1600 \times 24$ |
| Batch Normalization | $1600 \times 24$ |
| MaxPooling (stride 4) | $400 \times 24$ |
| Convolution (128 filters, size 3, stride 1) | $400 \times 128$ |
| Batch Normalization | $400 \times 128$ |
| MaxPooling (stride 4) | $100 \times 128$ |
| Convolution (256 filters, size 3, stride 1) | $100 \times 256$ |
| Batch Normalization | $100 \times 256$ |
| MaxPooling (stride 4) | $25 \times 256$ |
| Convolution (512 filters, size 3, stride 1) | $25 \times 512$ |
| Batch Normalization | $25 \times 512$ |
| MaxPooling (stride 4) | $6 \times 512$ |
| Global Average Pooling | $512$ |
| Dense 10 | $10$ |

Table 8: SampleCNN-SpectroBank basic block. Choice of $k$ is detailed in Kim et al. (2019)

| Layer | Output size |
|---|---|
| Input | $N \times d$ |
| Convolution ($k$ filters, size 4, stride 1) | $N \times k$ |
| Batch Normalization | $N \times k$ |
| MaxPooling (stride 2) | $\frac{N}{2} \times k$ |

Table 9: SampleCNN-SpectroBank architecture for Google Speech Commands ($n = 35$) or Urban-sound8K ($n = 10$). *BB* stands for 'Basic Block', and *GMP* for 'Global Max Pooling'

| Layer | Output size |
|---|---|
| Input | $16000 \times 1$ |
| SpectroBank (80 filters, size 160, stride 40) | $200 \times 80$ |
| Batch Normalization | $200 \times 80$ |
| BB 0 ($k = 80$) | $100 \times 80$ |
| BB 1 ($k = 80$) | $50 \times 80$ |
| BB 2 ($k = 160$) | $25 \times 160$ |
| BB 3 ($k = 160$) | $12 \times 160$ |
| BB 4 ($k = 160$) | $6 \times 160$ |
| BB 5 ($k = 160$) | $3 \times 160$ |
| BB 6 ($k = 320$) | $1 \times 320$ |
| Concatenate (GMP(BB 4), GMP(BB 5), GMP(BB 6)) | 640 |
| Dense | 640 |
| Batch Normalization | 640 |
| Dropout (0.25) | 640 |
| Dense $n$ | $n$ |

## C  DATASETS

The overall statistics for all the datasets used in the experiments is given in Table 10. In addition the distribution of spectral energy per class is provided on Fig. 5. Most of the speech energy is located in the 0-1.5kHz band.

Table 10: Class statistics over different datasets.

| Database | # of samples | # of classes | largest class size | smallest class size |
|---|---|---|---|---|
| AudioMNIST | 30000 | 10 | 3000 | 3000 |
| GoogleSpeechCommands | 105829 | 35 | 4052 | 1557 |
| UrbanSound8K | 9732 | 10 | 1000 | 374 |

## D  ERB AND BARK SCALES

Two main models of auditory filter bank system provide the expression of a filter bandwidth $B$ with respect to its frequency $f$. In the Bark model (Zwicker & Terhardt, 1980) the expression is the following:

$$B_b(f) = 25 + 75[1 + 1.4 \left(\frac{f}{1000}\right)^2]^{0.69}, \qquad (6)$$

and in the ERB scale (Glasberg & Moore, 1990):

$$B_{ERB}(f) = 24.7(4.37f + 1). \qquad (7)$$

These expression are the ones used in the present work.

In addition, these auditory models provide expressions for the frequency spacing between consecutive filters, that follow a logarithmic law. For a given filter number $k$ in the set of filters, its frequency can be obtained by using the following formula: $f = 228.846 \left(e^{k_{ERB}/9.265} - 1\right)$. This relationship is more often expressed in terms of $k$ as a function of the frequency:

$$k_{ERB} = 9.265 \log \left(1 + \frac{f}{228.846}\right), \qquad (8)$$

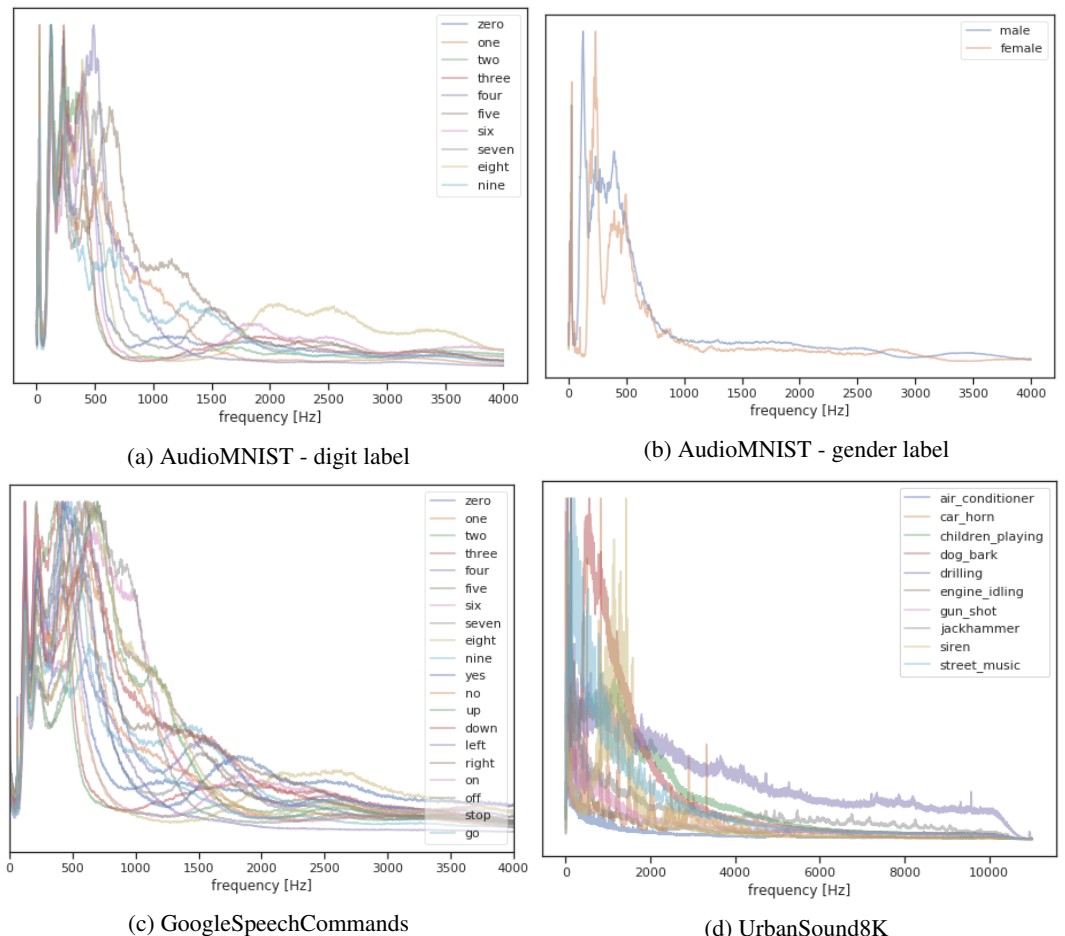

(a) AudioMNIST - digit label

(b) AudioMNIST - gender label

(c) GoogleSpeechCommands

(d) UrbanSound8K

Figure 5: Dataset energy distribution per class and corresponding labels.

The Bark model has a similar expression:

$$k_b = 13 \arctan\left(0.76\frac{f}{1000}\right) + 3.5 \arctan\left(\frac{f}{7500}\right)^2. \tag{9}$$

One can also compare with the Mel-scale. Sampling linearly on the Mel-scale $m$ leads to logarithmic frequency sampling:

$$m = 1127 \ln\left(1 + \frac{f}{700}\right). \tag{10}$$

