# OpenReview forum: "SpectroBank: A filter-bank convolutional layer for CNN-based audio applications"
_ICLR.cc/2020/Conference — Reject_

### Official Review · AnonReviewer1 · 2019-10-22
**Official Blind Review #1**

**Rating:** 3

**Review:**

The paper proposes a new type of convolutional layer for (band-pass) filtering of input signals (e.g., audio recordings). The main benefit is that the layer can be specified with a small number of parameters (+ filter configurations that are typically fixed beforehand get to be tuned to improve inductive bias). This is achieved via modulated windows or wavelets. While this is interesting, I do not see any conceptual novelty. Namely, previous work has already proposed and considered such layers. More specifically, in [1] the authors have considered exponentially-modulated Gaussian windows (detailed experiments, influence of different initialization strategies, properties of learned filters, distribution of modulation frequencies etc.). In [2] the layer is realized using wavelets. In [3] the filter is expressed as a difference between two sinc functions. The authors might argue that the conceptual difference compared to [1] is cosine modulation (see Remark 2 on page 4). Well, cosine modulated filters were considered in [4] as Parzen filters (v1 was on arXiv in June 2019). The latter work has not even been cited by the authors. Moreover, the paper does not discuss the consequences of using cosine modulations instead of exponentials. Section 2.2 in [4] explains why the use of cosine modulations is well suited for real-valued signals. In particular, the moduli of Fourier coefficients are symmetric around the origin for real-valued signals and for this reason spectrograms are typically computed over positive frequencies only. Thus, from this perspective it does not make much difference whether one uses cosine or exponential modulation (when it comes to standard feature extraction approaches for speech processing).

In the empirical evaluation the focus is on showing the utility of filter optimization on different tasks. The first experiments investigates basic properties such as how the number of filters and their overlap influence the effectiveness of a model. It is unclear why a single learning task is sufficient to conclude that more than 30 filters does not amount to an improvement in accuracy (128 and 64 filters are used in [3] and [4], respectively). This lack of reference to findings in previous work make the analysis incomplete. The approach is evaluated in total on three datasets: audio-mnist, google speech command, and urban-sound. While the reported results indicate a good performance of the considered approach over different tasks, the experiments completely ignore previous approaches for filter learning. This lack of baselines and reference to related work makes the experiments inadequate.

In general, my main concern with the experiments is that the section is written as if this is the first work proposing filter learning. I feel that a comparison to at least on of the baselines [1-4] would be required for a non-trivial assessment of the approach.


[1] N. Zeghidour, N. Usunier, I. Kokkinos, T. Schatz, G. Synnaeve, and E. Dupoux (ICASSP 2018). Learning filterbanks from raw speech for phone recognition.
[2] H. Khan and B. Yener (NIPS 2018). Learning filter widths of spectral decompositions with wavelets.
[3] M. Ravanelli and Y. Bengio (arXiv:1812.05920 2018). Speech and speaker recognition from raw waveform with SincNet.
[4] D. Oglic, Z. Cvetkovic, P. Sollich (arXiv:1906.09526 2019). Bayesian Parznets for Robust Speech Recognition in the Waveform Domain.


**Experience Assessment:**

I have read many papers in this area.

**Review Assessment: Checking Correctness Of Derivations And Theory:**

I did not assess the derivations or theory.

**Review Assessment: Checking Correctness Of Experiments:**

I assessed the sensibility of the experiments.

**Review Assessment: Thoroughness In Paper Reading:**

I read the paper at least twice and used my best judgement in assessing the paper.

---

> ### Author Response · Authors · 2019-11-12
> **Response to comments from reviewer #1**
>
> Thank you for your comments.
> We would like to address your response in detail, and try to clarify the misunderstandings. As can be seen at the first glimpse over the list of cited literature, we have cited papers [1-3] within the Motivation and Related Work sections and are familiar with the details of these papers:
>
> In paper [1] authors have used the initialization on a Mel scale and then trained the filter in a non-parametric way. Unlike in their paper, we have tried to show that this type of initialization is not necessary, and that parametric filters learned converge to the Bark scale. So, the novelty is twofold, since we have gone beyond showing the convergence trends for the frequencies, but also for the corresponding bandwidth. In most papers having studied parametric filter banks, filter initialization is done following a psychoacoustic scale or using prior knowledge about the signal under consideration. While this can speed up training times, this is an unnecessary step.
>
> In paper [2] authors have used single parameter filters in a filter bank, which are insufficiently adaptable to the task at hand, since the frequency-bandwidth relationship in Wavelets is not adapted to the perceptual models in audio applications. Running the AudioMNIST experiment using the learnable Wavelet filter bank in the ‘SpectroBank-Audionet’ from [2] gives 89.9%+-1.18% accuracy which is much lower than both the baseline and our experiment using a Gammatone learnable filterbank. When using the learnable Wavelet filter bank on the simplified Spectrobank network, accuracy is even worse, dropping to 88.9 % +- 1.43%. It is also the case for the experiments performed on GoogleSpeechCommand dataset, varying overlap ( see Fig. 1 (a)) the Wavelet filter-bank yields the worst results.
>
> In paper [3] authors have used sinc parameterized filters. Using a SincNet first layer (with Mel-scale pre-initialization) in Spectrobank-Audionet yields an accuracy of 97.0% +- 0.5%, which is very close to the results presented in the paper, using learnable Gammatones. When using the simplified Spectrobank network, accuracy is 97.2%+-1%, also close to the performance of learned Gammatones. The Mel-based filter initialization used (also in [1]) by Ravanelli et al. has negligible impact on the results.
> In order to better show the impact of the first layer, we did another AudioMNIST experiment with a much simpler network, having only 50k trainable parameters. This very simple model architecture is made of the following layers:
> - Spectrobank
> - Maxpooling (4, stride=4)
> - Dense (16) + Dropout(0.5)
> - Softmax output layer (10 classes)
> Again with this setting, differences proved to be quite small between the two settings: 79.9%+-4.3% for Gammatones and 80.6%+- 4% for SincNet.
> In conclusion is that plugging SincNet as a learnable filter shows performance that fits into the observations made in Fig. 1a, showing very close results for Gammatone/Gammachirp/Gaussian filter banks as first layer, and we will add the SincNet accuracy curve on Fig. 1a in the revised version of our paper.
>
> We did not claim that cosine modulation was the novelty in our paper. This is just a way of simplifying implementation and dealing with real values instead of complex ones. Thanks nevertheless for bringing to our knowledge reference [4] which is however only marginally relevant to our work.
> We would like to put the emphasis on the fact that our paper was addressing the question of convergence of parametric filter banks to the perceptual scale, without prior initialization using known perceptual models.
>
> Regarding the influence of the number of filters present in the filter bank, our findings with respect to a very specific classification task (GoogleSpeechCommand) indicate that the ‘sweet spot’ is close to 32 filters. However, when performing another task (e.g. source separation), it can be expected that this optimal value becomes larger.

---

### Official Review · AnonReviewer2 · 2019-10-23
**Official Blind Review #2**

**Rating:** 3

**Review:**

This paper proposes to specify the first layer of a CNN for audio applications with predefined filterbanks from the signal processing community. Those latter are only specified by a limited number of parameters, such as the bandwidth or the central frequency of the filter, and those parameters are then optimized through the standard back-propagation algorithm. Some accuracy improvements are obtained on non trivial datasets.

I think there are a lot of interesting ideas and the numerical improvements seem consistent with the method. However, I find that this study would benefit of more careful comparisons to understand which particular component is responsible for some of their success! Also, I think some relevant papers are missing in the introduction.


Pros :
- Good numerical performances.
- Interesting study of the impact of predefined filters; an analysis at the end of the paper(bandwidth, principal frequency chosen by the algorithm) is shown, which is a positive aspect of the paper.

Cons :
- Several attempts to employ hybrid architectures (as defined in the text) have been already proposed. References to hybrid architecture from Mallat's group are missing, e.g.:  https://arxiv.org/abs/1809.06367/ https://arxiv.org/abs/1605.06644 . Another line of work concerns the steerable filters, which is another manner to parametrized the filters and learn them (e.g.: https://www.cv-foundation.org/openaccess/content_cvpr_2016/papers/Jacobsen_Structured_Receptive_Fields_CVPR_2016_paper.pdf ) Another manner could be to directly learn the filters as wavelets: https://arxiv.org/pdf/1811.06115.pdf . I agree some of those references are only considering images, but those methods are definitely not specific to them.
- Comparing the number of parameters of hybrid and non hybrid architectures is meaningless in this setting, as in all the experiments, the number of parameters of the layers above the first layer are kept identical: one only sees the difference due to the first layer, whose kernel is indeed relatively high-dimensional.
- Also, my understanding is that the general pipeline is slower: indeed, a parameter update aims to compute @f/@w=@f/@x*@x/@w. The computation of the term @f/@x is unavoidable and is identical to its non-hybrid counter-part. However, @x/@w might be sometimes higher because the computations can involve potentially more complex functions(e.g., exponential, cos, sin contrary to linear functions). Would you mind to clarify this thought?(A small fair timing comparison would be welcome!)
- Furthermore, the improvement in performances is clearly thanks to those a-priori incorporated. As stated in the text, many works propose to initialize the CNN with a specific filter bank. Have the authors tried to compare their performances if the first layer is simply initialized with those filters and then freely evolve? I feel this is missing and would make the claim of the paper stronger. If this has been already done, please highlight it in the text.
- Abstract: it is claimed that this technique leads to a training speedup (i.e., less epochs) but I do not understand where this is shown.
- Section 3: Sometimes(e.g., AudioMnist), the hybrid training pipeline is quite different from the original implementation, for instance, because of the use of ADAM when the original implementation was using SGD. Did the use of a different optimizer affect the performances?(e.g., SGD?)
- Section 3.4: Why not comparing with data augmented settings?

Suggestions of improvement:
- I would have liked to see a littlewood-paley plot (eg, the sum of the modulus of the filters in the frequency domain) to understand better the distribution of the filters in the Fourier domain, in particular w.r.t. the high-frequency.
- "the output maybe too redundant"(page 5) - I don't understand why this would be an issue? In this case, the network should decide which coefficients to discard if the classifier is good enough, shouldn't it?

Post-discussion:
R1 made several relevant comments about the technical novelty and my concerns weren't fully solved. Thus I decided to maintain my score.

**Experience Assessment:**

I have published in this field for several years.

**Review Assessment: Checking Correctness Of Derivations And Theory:**

I carefully checked the derivations and theory.

**Review Assessment: Checking Correctness Of Experiments:**

I carefully checked the experiments.

**Review Assessment: Thoroughness In Paper Reading:**

I read the paper thoroughly.

---

> ### Author Response · Authors · 2019-11-14
> **response to reviewer #2 comments (1/2)**
>
> Thanks for your comments, please find below the answers to the points your raised.
>
> The first part of the response mentions Wavelets and the scattering transform. We will add the proposed papers to the state of the art section, discussing why our work is different. Our paper focuses on learning parametrized functions and analyzing the learned parameters, to get some insights about the data and learning. The scattering transform makes use, in each layer, of a set of wavelets with fixed scale (not learnable).
> As stated in [1], section 2.3: “Scattering uses a multi-layer cascade of a pre-defined wavelet filter bank with nonlinearity and pooling operators.“ We do not use pre-defined wavelet filter banks, the filterbank is learnt through the learnable parameters. The approach we have (same approach as in the papers we cite) is half way between a fixed filter bank and free learnable kernels. Again in section 2.3 of the paper cited above: “In contrast to Scattering, we learn linear combinations of a filter basis into effective filters and non-linear combinations thereof.“ They learn linear combination of a filter basis while we learn the filters and their combinations. The set of learned filters may be a basis or not.
>
>
> A more closely related work is the one of Khan et al (Neurips 2018) about learning parametrized Wavelets. We refer to it and use it as a baseline. However, we point out that Khan et al. do not cite any work on the scattering transform. This is unfortunate and we will add a reference to Mallat’s team work on the scattering transform.
>
> The purpose of the comparison of the number of parameters was included in the paper as  means of illustrating that  our layer  can enable training of smaller networks that will exceed the accuracy of the fixed (but more complex) representation networks.
>
> The choice of a specific initialization has been investigated and we showed (cf. section 3.5) that the training process moves the learned filters toward a perceptual scale. Choosing specific parameters might speed up the training but does not affect overall accuracy. While performing tests with SincNet to address reviewer 1 concerns, we replaced the mel-based initialization from Ravanelli et al. by the one used in Spectrobanks and did not observe any significant difference in overall accuracy.
>
>
> [1] Jacobsen et al. , Structured Receptive Fields in CNNs
> https://www.cv-foundation.org/openaccess/content_cvpr_2016/papers/Jacobsen_Structured_Receptive_Fields_CVPR_2016_paper.pdf

---

> > ### Comment · AnonReviewer2 · 2019-11-14
> > **Thanks for some clarifications**
> >
> > Dear authors,
> >
> > Thanks for your reply. I answer paragraph per paragraph. Only a portion of my comments have been addressed.
> >
> > 1/2/ Thanks for the clarifications. I agree with your statements, my point was more to help enriching the related works part.
> >
> > 3/ Well, "smaller" in which sens? If I'm correct, the kernel sizes are similar yet the parametrization rely on less parameters. What about the speed?
> >
> > 3/ Has this comparison been done systematically through the section 3.2, 3.3 and 3.4?(I saw a paragraph about the initialization but my understanding is that, for each experiment, it applies only to the original implementation on which the experiment is based on) If I'm correct, then, that could be highlighted more, and not only on the specific example of the SincNet.

---

> > > ### Author Response · Authors · 2019-11-14
> > > **addtional clarifications**
> > >
> > > Thank you for the quick feedback.
> > >
> > > 3) By smaller we mean "having less trainable parameters": smaller final dense layers and/or less intermediate convolution layers.
> > >
> > > 4) Due to timing constraint and comments , we focused on SincNet only. We did not have time to reproduce all experiments with SincNet, only the AudioMNIST ones. However SincNet does not benefit specifically form the mel-scale initialization as accuracy remains very close from either a learnt Gammatone initialized with linear frequency scale or even with a SincNet modified with initialization also with a linear frequency scale. This is mentioned in the revised version.

---

> ### Author Response · Authors · 2019-11-14
> **response to reviewer #2 (2/2)**
>
> The choice of the optimizer: indeed a change of the optimizer affects the performances and we selected the one giving the best results for each network we trained. That is why we get a better accuracy for some of the networks, compared to the original paper where they were presented. The accuracy numbers reported for the re-implementation of AudioNet for AudioMNIST were higher than the ones presented in the original paper (92.5% with SGD in the original AudioMNIST paper, vs. 94.9% with Adam in our re-implementation), providing a more fair comparison between the original Audionet and the Spectrobank-enabled Audionet.
>
> The number of filters is not of high relevance and is not a critical point of the solution, but we just wanted to observe the effect it has on accuracy. As mentioned in reviewer #1 response, the ‘optimal’ number of filters for the classification tasks we studied might prove to be quite different for other types of tasks.
>
> Regarding the faster training performance claim, when training Audionet, validation accuracy is greater than 93% for the first time after 13 epochs (and suffers from accuracy drops later) whereas when training Spectrobank-AudioNet, the 93% validation accuracy is reached after 4 epochs only (and does not become lower in the following epochs), but due to the non-negligible network settings, comparing convergence speed fairly is difficult.
>
> It is however true that despite its lower number of parameters, a spectrobank layer might slower than a non-parametric convolution layer, since you must generate the filters from parameters, and then perform convolution with longer filters. In our experiments based on a modified SampleCNN, an epoch of Spectrobank-enabled network was 1.5 times slower (3 ms / step vs. 2 ms / step) than the non-Spectrobank equivalent (same number of filters in the first layer, same batch size). However, the number of filters in the first layer needed by a non-parametric layer is much larger than in the parametric case to achieve the same (or better) results. SampleCNN’s 1st layer uses stride=3 which corresponds to a 98% overlap for a 10 ms filter. Our results shown in the paper use 75% overlap, corresponding to a case where stride=40.
>
> The overall speed gain in training we metion is partly caused by the reduced network architecture (less filters needed in the first layer) and quicker convergence of the learned parametric filters.
>
> In section 3.4, we agree that it would be more interesting to compare our results using the Urbansounds dataset with state of the art performance, requiring to use data augmentation. We will add those results in the final version of the paper (those experiments require additional work that we do not have time to perform within the rebuttal period).
>
> Concerning the discussion presented in page 5 regarding the 99% overlap, the results computed are in fact incorrect (due to a rounding error in the implementation), making the discussion of this particular case irrelevant. This will be updated in the revised version.

---

> > ### Comment · AnonReviewer2 · 2019-11-14
> > **Thanks!**
> >
> > I apologize for the former message, this reply solves most of my concern. I appreciate your honesty. I will reconsider my review. Thanks.
> >
> > I think the authors should find a precise setting to highlight the speed gain (convergence/individual layers), yet I agree this is a difficult task. If this is not possible, then such claims should be removed from the manuscript.
> >
> > I have the same thinking w.r.t. the initialization (stated above), for which some clarifications could help to understand better the improvements due to this method, in addition to the analysis of section 3.5.(I'm still not sure if each NNs of the sections 3.2-3.4 have been initialized with a "good" initialization or not)

---

### Official Review · AnonReviewer3 · 2019-10-24
**Official Blind Review #3**

**Rating:** 3

**Review:**

The paper presents an interesting signal processing-based extension of CNNs, where the first layer convolution is replaced by some pre-defined filter banks. Since those filter banks are parameterized with a smaller number of parameters, while they have been proven to be effective in audio processing, I was convinced that this approach could produce better performance than a generic CNN with no such consideration.

I am still wondering though, what is the main difference between this approach and Wavelet transform-based scatter transform networks that Stephane Mallat has proposed for years, for example in (Andén and Mallat 2014). I figure the proposed method in this paper is more flexible as it does not use the pre-defined filterbanks; instead it tries to learn the parameters to specify the only necessary filters for the particular problem. But I think the authors may need to address the difference from this previous work done by Mallat's group, because they at least share a similar philosophy.

Another thing that's not entirely clear for me was the effect of the filter length. Obviously, it should depend on the particular classification problem. For example, for speech, there needs to be consideration about the shortest stationary period of speech, while in some other cases like music and urban sound, it should be in different lengths to capture the specifics. It's a bit hard for me to believe that the different choices of filter banks from 1 to 100 ms all gave the same results (in Figure 1b). I think, if there is an optimal filter length depending on the problem, which has to be found to guarantee the performance, it has to be better investigated in the paper.

It is a confusing message to me, because the paper claims that the first layer of their network can cover a large area, which responds to a large receptive field, with a single filter by using a different parameter. It is a clearly a different kind of observation than the computer vision networks where the large receptive fields are defined with a deeper architecthre and strides. However, the shortest filter (1ms) and the longest one (100ms) doesn't make any difference, empirically? More discussion is needed to resolve this confusion.


J Andén and S Mallat, "Deep scattering spectrum", IEEE Transactions on Signal Processing, 2014

**Experience Assessment:**

I have published in this field for several years.

**Review Assessment: Checking Correctness Of Derivations And Theory:**

I assessed the sensibility of the derivations and theory.

**Review Assessment: Checking Correctness Of Experiments:**

I assessed the sensibility of the experiments.

**Review Assessment: Thoroughness In Paper Reading:**

I read the paper at least twice and used my best judgement in assessing the paper.

---

> ### Author Response · Authors · 2019-11-15
> **response to reviewer #3**
>
> Thank you for your comments. We apologize for the late answer, however addressing your comments required us to perform additional experiments.
>
> For the relationship with the work of Mallat’s group, thank you for your remark, please refer to the answer to reviewer #2 as he had similar remarks.
>
> The results presented in Figure 1b suffered from an implementation issue (namely a rounding error) which have now been corrected, as well as their analysis, in the revised version of the paper. However, due to the short time available to address all reviewers’ comments, we did not have the time to fully reproduce the experiment from Fig. 1b, as some settings require (especially 90% overlap) large training times.
>
> Influence of the filter length: Looking more carefully at the bandwidth of the learned filters, a majority of them have a bandwidth close to 100Hz or greater. If we take the example of the Gaussian filter where a width (variance) of sigma in the frequency domain corresponds to a width of 1/sigma in the time domain, we obtain a Gaussian window with a width (variance) of 10 ms. Accordingly, if the frequency bandwidth is higher, the time width is smaller. So that a filter size of 10ms is sufficient to fit most of our functions. Filter length of 100 ms would be useful to capture low frequency components (lower than 100Hz) but it seems that it is not necessary for the dataset we analyze within the scope of the task at hand, which is audio classification and in particular speech signals. Most of the relevant information for classification can be found in the part of the spectrum above 100Hz. That is why, in our setting, an increase of the filter size beyond 10 ms has no influence on the accuracy (assuming a stride of one sample).
> Concerning the influence of the overlap (stride) in the range of filter lengths 1 to 10ms, we have performed a new experiment to test it, with a corrected implementation. We replaced the plot (Fig. 1b) showing the evolution of the accuracy in this range. We observe an interesting behavior: the curves for the different overlap values cross in this range. We added the following text to the paper to explain the bad accuracy for large overlap with small kernel size < 4 ms: “On the other extreme, short kernels (less than 4ms) with large overlap (or small stride), can render the network short-sighted in time. In that case, long temporal patterns require the combination of a large amount of successive output values. The convolutional layers following the SpectroBank layer, deeper inside the network, may not be able capture these long patterns. This results as well in a drop of the accuracy observed on Fig. 1b.”
> Thank you very much for your careful reading and pointing the confusion in Fig. 1b. We were able to correct it and the results look (to us) much more logical.

---

> > ### Comment · AnonReviewer3 · 2019-11-15
> > **Acknowledgement of the rebuttal**
> >
> > The authors have cleared most of my technical concerns and lack of bibliography about the Wavelet-based scatter transform networks. It somewhat shows the merit of the learnable filterbanks in that the learned filter lengths are short unless they have to. However, while logically convincing, I still wish that there could be some experimental contrasts against the more deterministic methods, especially in terms of complexity.

---

### Decision · Program_Chairs · 2019-12-19

**Decision:**

Reject

**Comment:**

The paper proposed a parameterized convolution layer using predefined filterbanks. It has the benefit of less parameters to optimize and better interpretability. The original submission failed to inlcude many related work into the discussion which was addressed during the rebutal.

The main concerns for this paper is the limited novelty and insufficient experimental validation and comprisons:
* There have been existing work using sinc parameterized filters, learnable Gammatones etc, which are very similar to the proposed method. Also in the rebutal, the authors acknowledged that "We did not claim that cosine modulation was the novelty in our paper" and it is "just a way of simplifying implementation and dealing with real values instead of complex ones" and "addressing the question of convergence of parametric filter banks to perceptual scale".
* Although the authors addressed the missing related work problem by including them into discussions, the expeirmental sections need more work to include comparisons to those methods and also more validations on difference datasets to address the concern on the generalization of the proposed method.